# The Presence of *Exophiala dermatitidis* in the Respiratory Tract of Cystic Fibrosis Patients Accelerates Lung Function Decline: A Retrospective Review of Lung Function

**DOI:** 10.3390/jof8040376

**Published:** 2022-04-07

**Authors:** Jonathan Ayling-Smith, Lorraine Speight, Rishi Dhillon, Matthijs Backx, Philip Lewis White, Kerenza Hood, Jamie Duckers

**Affiliations:** 1University Hospital of Wales, Cardiff and Vale University Health Board, Cardiff CF14 4XW, UK; 2College of Biomedical and Life Sciences, Cardiff University, Cardiff CF10 3AT, UK; hoodk1@cardiff.ac.uk; 3All Wales Adult Cystic Fibrosis Centre, University Hospital Llandough, Penarth CF64 2XX, UK; lorraine.speight@wales.nhs.uk; 4Public Health Wales, Cardiff CF10 4BZ, UK; rishi.dhillon@wales.nhs.uk (R.D.); matthijs.backx2@wales.nhs.uk (M.B.); lewis.white@wales.nhs.uk (P.L.W.)

**Keywords:** cystic fibrosis, *Exophiala dermatitidis*, lung function

## Abstract

*Exophiala dermatitidis* is increasingly isolated from cystic fibrosis (CF) respiratory samples. The decision to treat is hampered by limited evidence demonstrating the clinical significance of isolating *E. dermatitidis*. The objective was to assess the impact of *E. dermatitidis* isolation on the lung function of CF patients. The rate of lung function decline in the local CF population was calculated using historic lung function data. A control population who had never had *E. dermatitidis* cultured from the respiratory tract was compared with the *E. dermatitidis* group, calculating their rate of lung function decline before and after the first isolation of the organism. A total of 1840 lung function measurements were reviewed between the 31 *E. dermatitidis* group patients and 62 control patients. Their demographics were similar. The control group declined at a rate of −0.824 FEV1%/year. The rate of decline in the *E. dermatitidis* group prior to infection was −0.337 FEV1%/year (*p* = 0.2). However, post infection with *E. dermatitidis*, there was a significant increase in the rate of decline in lung function (−1.824 FEV1%/year, *p* < 0.01). The results suggest *E. dermatitidis* has a temporal relationship with accelerated rate of lung function decline. It is not clear if this is a cause or effect, but this accelerated rate of decline indicates a need for further investigation.

## 1. Introduction

Patients with cystic fibrosis (CF) are at an increased risk of pulmonary colonisation by opportunistic microorganisms. These microorganisms include the thermophilic black yeast *Exophiala dermatitidis*, which is isolated in 1–16% of the bronchial secretions of patients with CF [1,2]. *E. dermatitidis* rarely causes infection in the immunocompetent patient, but is more prevalent in the immunocompromised and has been identified as both a commensal bystander and important cause of invasive disease [1]. Aside from superficial skin infection, the respiratory system is the most common site of *E. dermatitidis* infection [3]. Despite this, the pathological mechanism of respiratory infection remains unclear. In the CF lung in particular, it is likely that the nutrient-rich, heavy mucous burden within the lung, coupled with inadequate mucociliary clearance promotes fungal growth, whilst the ongoing use of broad-spectrum antimicrobials may positively select fungi through the stifling of the growth of their competitors [4]. However, the variability of morbidity associated with this organism in both the immunocompromised and immunocompetent patient suggest that there is a more complex interaction of virulence factors, risk factors, and host responses at play than wholly the pulmonary-specific CF defects [3]. It is likely that the mode of infection is through respiratory inhalation of this ubiquitous organism. Man-made humid and hot habitats have a far higher isolation rate than the natural environment, and therefore it has been further postulated that aerosolization of the organism through hot equipment such as dishwashers or saunas may be a route of transmission [1,5].

Despite the growing body of evidence of the importance of fungi, there is no current consensus on the clinical management of the CF patient where *E. dermatitidis* has been isolated from the respiratory tract [4]. When deciding to treat *E. dermatitidis*, it is important to balance the potential risk/benefits of treatment initiation. The azole family of antifungals is often selected as first line therapy but have considerable side effects and drug–drug interactions, including interactions with CFTR modulators, leading to a requirement for therapeutic drug monitoring. However, balancing the risk against the potential benefits of treatment is difficult when there is limited evidence on the clinical consequences of *E. dermatitidis* presence in the CF airways.

Despite treatment advances, lung disease remains the major cause of mortality and morbidity in CF, and strategies to minimize pulmonary exacerbations and maintain lung health remain of paramount importance. Percent predicted forced expiratory volume in one second (FEV1%) is routinely used by clinical teams and registries to monitor clinical stability, stratify disease severity and assess treatment outcomes [6]. The rate of lung function decline, determined by comparison of FEV1% values over time, is perhaps an even more relevant and better predictor of clinical deterioration [7]. Prior studies have suggested that *E. dermatitidis* does not contribute to lung function decline, but patient numbers were limited and the length of follow up was short [8,9]. The primary objective of our study was to assess if the presence of *E. dermatitidis* in respiratory cultures impacts the trajectory of lung function decline of CF patients, which would help better inform the decision whether to initiate treatment targeting *E. dermatitidis* in future patients.

## 2. Materials and Methods

A service evaluation in the form of a single-centre retrospective case–control study was performed using data gained as part of routine clinical practice with no impact on patient management. Data on adult patients with CF receiving care at the All Wales Adult CF Centre, from whom *E. dermatitidis* had been cultured from respiratory tract, were compiled from retrospective review of routinely collected clinical data to generate the case population. They were compared with a randomly selected control group of *E. dermatitidis*-negative CF patients managed over the same period at the same CF centre. The control group was chosen to be 2x case and potential confounding factors such as age, CF-genetics, body mass index (BMI), and pancreatic sufficiency status were recorded to ensure the two groups were comparable.

Their lung function data in the form of FEV1 was documented using clinic letters and local spirometry databases. With data collected over several years, calculations to derive the percentage-predicted FEV1 will likely have varied so the actual FEV1 was documented, and all percentage-predicted measurements were individually recalculated using the same algorithm. 

Patient data were included up to lung transplantation, starting CFTR modulator therapy (including clinical trials), moving out of area, or death. A historical limit of 2007 or one year before coming under the care of the adult CF team (whichever was later) was used as, prior to 2007, the availability of local lung function data is variable and would likely introduce significant paediatric data, making predicted FEV1 data unreliable [10]. Patients with less than 48 months of continuous data available were also excluded, to allow for 2 calculations of annual lung function decline per patient. 

All respiratory cultures conducted over the period of interest from both groups were reviewed in order to establish if growth of non-tuberculous Mycobacterium, *Aspergillus fumigatus* or *Pseudomonas aeruginosa* was documented and potentially associated with FEV1% decline. Patients were recorded as being positive or negative for these organisms at the start and end date of data collection and “point of infection” in the *E. dermatitidis* group. This positive status was determined by at least one isolation culture. The control group was subject to the same patient and data selection limitations as the *E. dermatitidis* group.

All of the data were collated in a password protected database stored on a hospital networked PC and analysed using Microsoft Excel 2016 and IBM SPSS statistics 26. 

The rate of decline in lung function in each patient prior to infection with *E. dermatitidis* was calculated as FEV1%/year by finding the difference between the mean of all FEV1% from the earliest 12 month period on record and the mean FEV1% from the most recent 12 month period, ending at the date *E. dermatitidis* was 1st isolated, divided by the time studied. A second “post-infection” rate of decline was calculated in the same patient in a similar manner using the means of the FEV1% measurements for the immediate 12 months after infection and the most recent 12 months, subject to the limitations described above (transplant, CFTR use, etc.). The control group rate of decline was calculated as FEV1%/year, documenting the difference between the mean of all FEV1% from the earliest 12-month period on record and the mean of the FEV1% from the most recent 12-month period, subject to the same limitations as before. This difference in each patient’s paired means was divided by the time studied (in months) multiplied by 12 to achieve a FEV1% change/year. The rates of change of all the patients were pooled by finding a mean decline to give a control group rate of decline.

Patients with a positive sputum sample for *E. dermatitidis* were compared with the control group. Clinical, demographic, and microbiological characteristics were compared. The *E. dermatitidis* group (pre infection) and the control group were compared to see if patients were similar in demographics and their lung function declined at the rate that was similar to the control group. The pre and post infection time points in the *E. dermatitidis* group were then compared to assess if they underwent a significant change after infection, including their rates of decline. Demographic, microbiological, and baseline clinical characteristic data were compared by way of *T*-tests for continuous variables, Chi-2 for binary variables, and Fisher’s exact for binary variables with less than 10 cases. Similar tests were used when comparing within the *E. dermatitidis* group pre and post infection, except a paired *T*-test was used for continuous variables and McNemar for binary variables.

## 3. Results

From the 306 adult CF patients managed by the CF unit over the given period, a total of 61 patients had *Exophiala dermatitidis* cultured from the respiratory tract. After exclusion criteria were applied, 31 *E. dermatitidis* positive patients remained, and subsequently the control comprised 62 patients. The control group and *E. dermatitidis* group were comparable in regard to gender, CF genotype, proportion that were pancreatic sufficient, and amount of time studied. The demographics of this are shown in Table 1 and Figure 1. There was a slight, but statistically significant, age difference between the two groups at the start of data collection.

A total of 7 patients (22.6%) had paediatric readings in the *E. dermatitidis* group. Removing them would have made the mean age 21.8, which would have been comparable to the control group (*p* = 0.242). However, the decision was made to keep these patients in the analysis, as reducing the sample size by 7 would have been statistically costly and the paediatric bias was mitigated as much as possible through the recalculation of the predicted FEV1% data.

A total of 1840 lung function measurements were reviewed, and predicted FEV1% recalculated, from the 62 control patients at a mean of 15 measurements per patient. The average time between the first measurement recorded and most recent (subject to exclusion criteria) was 102.6 months. A total of 884 lung function measurements were recorded for the 32 patients in the *E. dermatitidis* group, which is 28.5 per patient, over an average time period of 97.8 months. Pre-infection with *E. dermatitidis*, there were 365 measurements, at an mean of 11.7 per patient, over 51.9 months, and post-infection there were 519 measurements, at an mean of 16.7 per patient, over 45.9 months. 

The pooled rate of decline of the control group was −0.82%/year (S.D 1.36); this was not statistically different (*p* = 0.212) from the rate of decline in the *E. dermatitidis* group prior to positive culture (−0.34%/year, S.D 2.38). However, there was a significant difference (*p* = 0.003) in rate of decline between the *E. dermatitidis* group pre and post first isolation (−0.34%/year compared to −1.82%/year, S.D 2.35), seen in Figure 2. The rate of decline between the control group was also significantly different to the *E. dermatitidis* group post-infection (*p* = 0.011), seen in Table 2. This is also worse than the UK national average CF lung function decline.

The BMI measurements of the control group and the *E. dermatitidis* group were not significantly different at the start of the data collection, but there was a significant increase in the rate of decline of BMI post *E. dermatitidis* infection (*p* = 0.011). Further analysis in comparing the endpoint BMI of the two groups did not show a significant difference (23.0, S.D. 4.12 in control group compared to 21.9, S.D. 2.92 in the *E. dermatitidis* group).

The *E. dermatitidis* group and the control group were comparable in rates of positive respiratory culture for non-tuberculous Mycobacterium at the start (*p* = 0.434) and at the end (*p* = 0.326) of the data collection. This was also the case for *Pseudomonas aeruginosa* culture (*p* = 0.433 and *p* = 0.451). Rates of *Aspergillus fumigatus*-positive culture were comparable at the start of the data (*p* = 1.00). However, when comparing end data while rates of *A. fumigatus* increased in the control population, they increased significantly more in the *E. dermatitidis* group (*p* = 0.004). 

Eighteen control patients (29%) changed from Aspergillus culture negative to positive during the data review. In the *E. dermatitidis* group, 20 patients (64.5%) changed Aspergillus culture status, representing a significant increase in the recovery of Aspergillus (*p* = 0.001). Further analysis showed that, in the *E. dermatitidis* group, 12 (38.7%) changed Aspergillus status before *E. dermatitidis* growth, and 11 (35.5%) after they had grown *E. dermatitidis*, suggesting that the presence of *E. dermatitidis* in the respiratory tract does not promote the presence of Aspergillus (*p* = 0.157). 

Further subgroup analysis was undertaken to understand if Aspergillus positivity was driving the lung function decline. The rates of decline after *E. dermatitidis* infection in those with negative *A. fumigatus* culture (*n* = 6) were compared to patients with positive Aspergillus culture (*n* = 11). Patients with persistent recovery of Aspergillus were excluded. The amount of time measured in each group was not statistically different. The rates of decline were not significantly different, with a rate of −1.64 (SD 2.31) in the *A. fumigatus* positive subgroup and a rate of −1.71 (SD 2.33) in the negative subgroup (*p* = 0.92). 

Further subgroup analysis was performed to establish if the presence of antifungals affected the rate of decline. The *E. dermatitidis* group were evaluated for all antifungal prescriptions issued to them and divided into an antifungal negative group if they had never received a prescription for antifungals, or an antifungal group if they were issued with an antifungal at any time after testing positive for *E. dermatitidis* up to the end point of their data collection, subject to the prior limits of data. In the *E. dermatitidis* group, 14 individuals were given antifungals and their rate of decline was −1.47% per year (SD 3.26) over an average of 45.0 months (SD 19.1), 17 individuals did not receive antifungals and they declined at a rate of −1.77% per year (SD 3.19) over an average of 45.9 months (SD 18.7). Even though the rate of decline if given antifungals appears to be slower, there was no significant difference between the two groups (*p* = 0.73).

## 4. Discussion

The results indicate an increased rate of lung function decline after isolation of *E. dermatitidis* in the CF lung. At present, it is not clear if this enhanced rate of lung function decline is a cause or effect of isolating *E. dermatitidis*. In this current study, the relationship with *A. fumigatus* is unclear; the presence of *E. dermatitidis* within the respiratory tract appears to increase the recovery rates of *A. fumigatus*, but the presence of *A. fumigatus* was not necessarily associated with decline in lung function. Previous work suggesting that *E. dermatitidis* could be a colonising bystander of a dysregulated airway [1] could be supported by this study, but the temporal nature of the isolation and rate of decline makes this less likely. Previous research described an association between the recovery of *E. dermatitidis* and that of *Aspergillus fumigatus* [8]. This may be related to the hypothesis that increased use of broad-spectrum antibiotics in the unwell CF patient positively selects all fungi in the respiratory tract [2], and the presence of *A. fumigatus* could be a marker of CF disease progression [11]. 

However, the presence of *A.fumigatus* in the CF respiratory tract is unlikely to be solely responsible for the decline observed in this study. Individuals from whom *A.fumigatus* was recovered prior to the presence of *E. dermatitidis* did not show significant decline in lung function. The limited difference between rates of *Aspergillus* growth before and after *E. dermatitidis* isolation implies that *E. dermatitidis* itself does not make the individual more susceptible to *Aspergillus*. The significant difference in the recovery of *Aspergillus* in control and *E. dermatitidis* groups at the end of the study period implies that both *E. dermatitidis* and *A.fumigatus* are both markers of CF disease progression or have a synergistic affect. This is also supported by the antifungal subgroup analysis; that the *E. dermatitidis* group did not decline in lung function slower on antifungals suggests that *E. dermatitidis* presence is a signal of severe disease rather than driving the deterioration. However, the numbers of individuals included in the subgroup analysis was small. 

There is a significant signal that *E. dermatitidis* isolation may be associated with BMI decline, which is not translated into a significant difference in the end measurement of BMI, so is unlikely to be a further feature of CF disease progression but rather may be a confounder of the anorexia experienced by the unwell patient during exacerbation; either due to the infection itself of the antimicrobials they will be receiving. The prescription of antifungals was only measured in the *E. dermatitidis* group before and after isolation and, whilst the difference in decline was not significant, the numbers in this subgroup analysis was small. In addition, an increase in these drugs commonly associated with anorexia and nausea could explain the apparent temporary BMI decline that did not translate to significant weight loss.

This study reviews patients with an uncommon condition who are isolating a rare pathogen from their respiratory tract and, as such, the numbers in each group are lower in this single centre compared with other published works [8]. However, this study reviews historic lung function over a far longer timeframe than previously studied, which may account for the elucidation of a modest decline difference [8,9]. Despite the smaller numbers, the patients were well matched in each group; beyond the relationship with *A.fumigatus*, the only significant difference between the groups was age. This is likely to be a confounding result brought about by the necessity of gathering more retrospective data in the *E. dermatitidis* group in order calculate a decline in lung function before and after infection, as the minimum amount of data needed was greater in this group. As such, patients were more susceptible to exclusion in the *E. dermatitidis* group than the control. This is a potential source of bias. Whilst the proportion of delta-F508 homozygous patients is similar in each group, there is still a limitation in the comparability to national data.

When considering sample size, the rate at which *E. dermatitidis* is being detected locally is increasing, potentially due to increased awareness of the organism. While many patients were excluded from the study as insufficient time had elapsed following positive culture to effectively measure a rate of decline, within 2 years the number of patients that could be included in this study could be much higher. This could allow for future work to be powered for multivariate logistic regression modelling and better matching of the controls, including matching by other microorganisms present within the respiratory tract of the CF patient.

## 5. Conclusions

This study suggests that *E. dermatitidis* isolation from the respiratory tract of adults with CF has a temporal relationship with an increased rate of lung function decline (1.82% compared to 0.33% pa) that exceeds the control group of adult CF patients at the same centre (0.82%) and the previously reported UK average (1.5%) [12]. Evidence indicates that *E. dermatitidis* is an organism of significance in the CF lung; whether it plays an active or passive role in the lung microbiota, it represents a priority signal to the CF clinician, particularly considering the importance of preserving lung function in CF [13]. Given the limited population studied, the evidence generated is not sufficient to support clinical decisions, such as antifungal use following isolation of *E. dermatitidis*, and performing larger, confirmatory studies would be a significant contributory step to resolving this clinical dilemma. 

## Figures and Tables

**Figure 1 jof-08-00376-f001:**
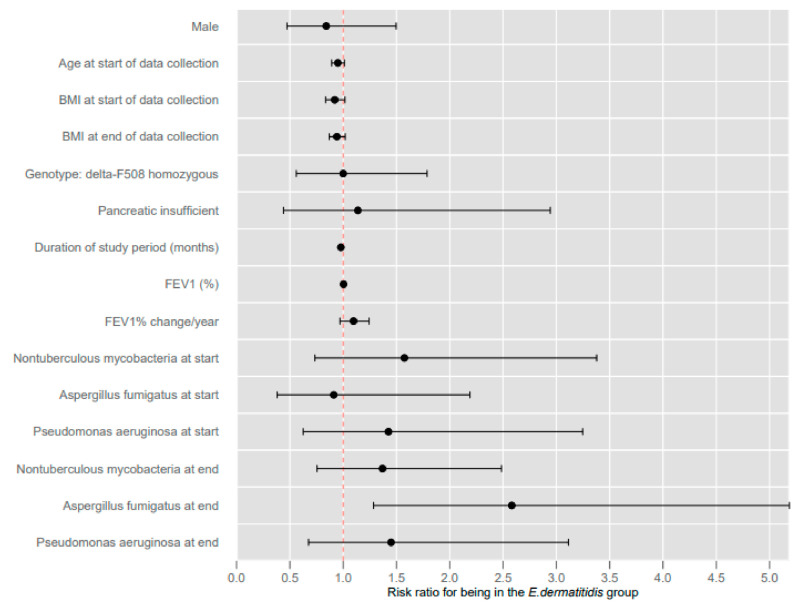
Clinical factors associated with *E. dermatitidis* group.

**Figure 2 jof-08-00376-f002:**
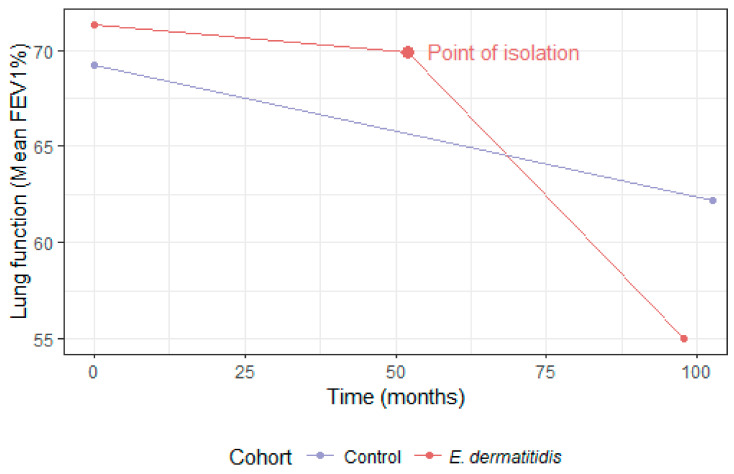
Lung function decline before and after infection in *E. dermatitidis* group compared with lung function decline in control group.

**Table 1 jof-08-00376-t001:** Clinical characteristics of *E. dermatitidis* group and Control group.

Parameter	Population		
	*E. dermatitidis* Group (*n* = 31)	Control Group (*n* = 62)	*p* Value
Sex, male (N, %)	16 (52)	36 (58)	0.555
Mean age at start of data collection (SD)	20 (7.92)	25 (10.5)	0.033
Mean age at date of positivity (SD)	24 (9.24)		
Mean BMI at start of data collection kg/m^2^ (SD)	20.8 (2.85)	22.0 (3.30)	0.113
Mean BMI at end of data collection kg/m^2^ (SD)	21.9 (2.92)	23.0 (4.12)	0.167
Genotype: delta-F508 homozygous (N, %)	14 (45.1)	28 (45.1)	1
Pancreatic insufficient (N, %)	28 (90.3)	57 (91.9)	0.79
Mean study period, months (SD)	97.8 (33.5)	102.6 (41.2)	0.58
Mean FEV1% at start of data collection (SD)	71.4 (17.5)	69.2 (21.3)	0.629
Mean FEV1% change/year (SD)	−0.34 (2.38)	−0.82 (1.36)	0.212
Microorganisms at start of data			
Nontuberculous mycobacteria N (%)	4 (12.9)	4 (6.5)	0.43
*Aspergillus fumigatus* N (%)	4 (12.9)	9 (14.5)	1
*Pseudomonas aeruginosa* N (%)	26 (83.9)	47 (75.8)	0.43
Microorganisms at end of data			
Nontuberculous mycobacteria N (%)	10 (32.2)	14 (22.6)	0.33
*Aspergillus fumigatus* N (%)	23 (74.2)	26 (41.9)	0.004
*Pseudomonas aeruginosa* N (%)	25 (80.6)	44 (71.0)	0.45

**Table 2 jof-08-00376-t002:** Clinical characteristics of the *E. dermatitidis* group before and after isolation.

Parameter	*E. dermatitidis* Group (Pre-Infection) (*n* = 31)	*E. dermatitidis* Group (Post-Infection) (*n* = 31)	*p* Value
Mean study period per patient in Months (SD)	51.9 (29.5)	45.9 (18.7)	0.36
Mean BMI change per year kg/m^2^/year (SD)	0.34 (1.75)	−0.02 (0.44)	0.011
Mean FEV1% change per year (SD)	−0.34 (2.38)	−1.82 (2.35)	0.005
Microorganisms			
Nontuberculous mycobacteria N (%)	4 (6.5)	10 (32.2)	0.250
*Aspergillus fumigatus* N (%)	9 (14.5)	23 (74.2)	0.022
*Pseudomonas aeruginosa* N (%)	47 (75.8)	25 (80.1)	1.00

## Data Availability

Not applicable. As an NHS service evaluation, the dataset cannot be made publicly available.

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
