# Peer review of "The Presence of Exophiala dermatitidis in the Respiratory Tract of Cystic Fibrosis Patients Accelerates Lung Function Decline: A Retrospective Review of Lung Function"

_jof, 2022, doi:10.3390/jof8040376_

Round 1

Reviewer 1 Report

The manuscript described the possibility of lung function decline by Exophiala dermatitidis. The authors showed that E. dermatitidis had a temporal relationship with accelerated rate of lung function decline, based on a retrospective review of lung function. Thus, these findings will be useful for the treatment of lung diseases. Therefore, the manuscript is not too excellent to be published. In other words, the manuscript is so excellent that it should be published.

Comments

(1) Is Exophiala dermatitidis commonly in lung? In what body situations is Exophiala dermatitidis in lung?

(2) How was human lung infected by Exophiala dermatitidis?

(3) What shall we do against Exophiala dermatitidis infection in the future?

(4) How is the correlation between cystic fibrosis and Exophiala dermatitidis?

(5) Is Exophiala dermatitidis in normal healthy persons?

That is all.

Author Response

Thank you for your review and comments.

  1. Is Exophiala dermatitidis commonly in lung? In what body situations is Exophiala dermatitidis in lung?
    • Line 30-31and 34-35 edited to answer this question about organism background
  2. How was human lung infected by Exophiala dermatitidis?
    • Line 35-45 edited to describe mode and route of infection
  3. What shall we do against Exophiala dermatitidis infection in the future?
    • 282-290 conclusion statement altered to describe better the future direction of research. The study alone is not able to support clinical decisions such as antifungal use with regards to E.dermatitidis
  4. How is the correlation between cystic fibrosis and Exophiala dermatitidis?
    • This is ultimately not clear in the literature but a summary of current most likely hypothesis given in Line 36-38 in conjunction with previous comments to better describe the relationship between CF and E.dermatitidis
  5. Is Exophiala dermatitidis in normal healthy persons?
    • Line 35 edited to describe E.dermatitidis infection in the healthy individual as rare.

Reviewer 2 Report

The study “The presence of Exophiala dermatitidis in the respiratory tract of cystic fibrosis patients accelerates lung function decline: a retrospective review of lung function” gives information about the FEV1 and Ed status in a cohort of 31 CF patients with Ed and a control group with 62 CF patients.

The study is of interest and novelty. The methods are sufficiently described. Data presentation is ok. However, I have some minor things that need to be improved.

  • E. dermatitidis is often not written in italic
  • Please add in Line 31 also the current review of Kirchhoff et al. DOI: 10.1080/21505594.2019.1596504 as reference for the prevalence of Ed in CF.
  • Was an ethical statement necessary?
  • What were the definitions for isolation of A. fumigatus or P. aeruginosa? Leeds criteria? Single isolation?
  • Quality of Figure 1 needs to be improved?
  • 2x Figure 1 is in the ms
  • In Figure 2 spelling of Ed needs correction
  • Conclusion is not a conclusion. It’s a more repetition and includes also new facts like the pediatric patients. Please change.

Author Response

Thank you for your review and your comments. These have been addressed in the following manner.  

  • E. dermatitidis is often not written in italic
    • References to E.dermatitidis have been italicised. All organisms in tables and graphs have also been italicised
  • Please add in Line 31 also the current review of Kirchhoff et al. DOI: 10.1080/21505594.2019.1596504 as reference for the prevalence of Ed in CF.
    • Thankyou for this suggestion. This reference has been added to Line 31
  • Was an ethical statement necessary?
    • It was not felt that a separate ethical statement was necessary. An informed consent statement was included describing this work as a retrospective service evaluation of existing data obtained during routine care, with the main objective to identify whether the presence of E. dermatitidis in the lung of the CF patient was clinically significant, potentially requiring attention when managing future CF patients (I.e. a development of routine service). This potential change to our routine clinical protocol was not generalizable to other populations. There were no other ethical issues that we felt warranted a separate statement.
  • What were the definitions for isolation of A. fumigatus or P. aeruginosa? Leeds criteria? Single isolation?
    • A single isolation of A.fumigatus and P.aeruginosa was enough for them to be categorised as being positive for this organism. This has been made clearer in the methods. (Line 89)
  • Quality of Figure 1 needs to be improved?
    • That which was erroreously labelled as figure 1 has been updated (Line 152)
  • 2x Figure 1 is in the ms
    • Altered to Figure 1 and Figure 2 appropriately (Line 166)
  • In Figure 2 spelling of Ed needs correction
    • This graph has now been altered (Line 170)
  • Conclusion is not a conclusion. It’s a more repetition and includes also new facts like the pediatric patients. Please change.
    • Comments regarding paediatric patient data has been altered moved to the results to make more narrative sense (Line 141-146). Multiple areas of the conclusion removed and points regarding limitations added to the discussion without repetition (Line 270-272, 221-224. Conclusion altered to better reflect a conclusion with a focus on looking forward to future work and its impact (Line 274-284)

Round 2

Reviewer 2 Report

All points were addressed accordingly.